# Laboratory study of non-linear wave-wave interactions of extreme focused waves in the nearshore zone

Iskander Abroug [1, 2], Nizar Abcha [1], Armelle Jarno [2], François Marin [2]

[1] Normandie Université, UNICAEN, UNIROUEN, CNRS, UMR 6143 M2C, 14000 Caen, France.
[2] Normandie Université, UNILEHAVRE, CNRS, UMR 6294 LOMC, 76600 Le Havre, France.

*Correspondence to*: Iskander Abroug (iskander.abroug@unicaen.fr)

**Abstract.** Extreme waves play a crucial role in marine inundation hazards and coastal erosion. Prediction of non-linear wave-wave interactions is crucial in assessing the propagation of shallow water extreme waves in coastal regions. In this article, we experimentally study non-linear wave-wave interactions of large amplitude focused wave groups propagating in a two-dimensional wave flume over a mild slope ($\beta$ = 1:25). The influence of the frequency spectrum and the steepness on the nonlinear interactions of focused waves are examined. The generated wave trains correspond to Pierson-Moskowitz and JONSWAP ($\gamma$ = 3.3 or $\gamma$ = 7) spectra. Subsequently, we experimentally approach this problem by the use of a bispectral analysis applied on short time series, via the wavelet-based bicoherence parameter, which identifies and quantifies the phase coupling resulting from non-resonant or bound triad interactions with the peak frequency. The bispectral analysis shows that the phase coupling increases gradually and approaches 1 just prior to breaking, accordingly with the spectrum broadening and the energy increase in high frequency components. Downstream breaking, the values of phase coupling between the peak frequency and its higher harmonics decrease drastically and the bicoherence spectrum becomes less structured.

## 1 Introduction

Extreme wave propagation is a highly nonlinear process observed in both open seas and coastal regions. The main physical mechanisms which may lead to an extreme wave event are illustrated in Kharif and Pelinovsky, 2003; Kharif et al., 2009; Didenkulova et al., 2010; Onorato et al., 2013. Extreme waves may occur in deep or shallow water, in energetic storm sea state or previously calm sea state. In our opinion, spatio-temporal wave focusing is one of the most important mechanisms in the extreme wave formation for shallow and deep water (Kharif and Pelinovsky, 2003). The spatio-temporal wave focusing is a classical mechanism giving rise to an important wave energy concentration in a small region. If the wave height of the focusing group exceeds 2.2 times its significant wave height, it can be defined as a rogue or freak wave (Dysthe et al., 2008). For this reason, spatio-temporal wave focusing is often employed in laboratory wave flumes with a wide variation of water depth (Merkoune et al., 2013), spectrum type (Tian et al., 2011; Xu et al., 2019; Abroug et al., 2019; 2020) and wavelength to depth ratio, in order to better understand the generation process, the dynamic behavior and the hydrodynamic loads on ocean structures in extreme sea conditions.

Over the past years, several studies have attempted to quantify the spatial evolution of spectral energy of unidirectional wave groups in experimental wave flumes using a classic Fourier analysis (Tian et al., 2011; Liang et al., 2017; Abroug et al., 2020). The frequency spectrum only gives the distribution of energy in the frequency domain; however, information about the phase coupling between different wave components is unknown. Consequently, higher order spectrum techniques should be adopted. A powerful tool to investigate the highly nonlinear process is the wavelet-based bispectral technique, which has been used in

several works to study the non-linear interactions and quadratic phase coupling between wave components (Dong et al., 2008; Ma et al., 2010). The need to detect and quantify second order non-linear interactions can be found in many disciplines, such as, geophysics (Grinsted et al., 2004), plasma physics (Milligan et al., 1995), fault diagnosis (Li et al., 2014), health-related areas, neuroscience (Bai et al., 2017) and wave analysis (Eldeberky, 1996; Eldeberky and Madsen, 1999; Young et al., 1996; Young and Eldeberky, 1998; Becq-Girard et al., 1999; Huseni and Balaji, 2017; Zhang et al., 2019). In wave analysis, the

propagation of wave trains in the nearshore zone has an exceptionally high spectral and temporal resolution.

The majority of previous works regarding the evolution of unidirectional wave trains in numerical and experimental wave flumes have shown that spatio-temporal focusing leads to a shape and elevation of a wave crest at focus that cannot be predicted by either linear or $2^{nd}$ order wave theory. This is due to high order nonlinearities, called the bound (harmonics) and resonant nonlinearities (Vyzikas et al., 2018). On the one hand, bound nonlinearities are the result of non-linear harmonics that are

phase-locked to the wave train and contribute in the sharpening of free surface elevation. On the other hand, resonant interactions contribute in the redistribution of energy among different frequency components. It is important to mention here that in shallow water regions exact resonant interactions are hardly realised in unidirectional propagation because the resonant conditions cannot not be satisfied in a small area. Therefore, we investigate specifically the role of bound waves generated by non-resonant three-wave coupling.

Over the past few decades, various experimental studies have investigated the spatial evolution of non-linear coupling between wave components. Dong et al., 2008 studied the spatial evolution of non-linear interactions between different wave components in the shoaling and de-shoaling region by carrying out two random waves experiments based on JONSWAP spectra with varying peak waves periods and root-mean-square waves heights. They showed that the degree of quadratic phase coupling increases in the shoaling region and achieves its highest level prior to wave breaking. Ma et al., 2010 studied

experimentally JONSWAP wave trains propagating in intermediate water depth. Recently, nonlinear transformation of unidirectional irregular waves propagating over a complex bathymetry ($1.06 < k_p h < 2.2$; where $k_p$ is the peak wavenumber and $h$ denotes the water depth) was performed in Zhang et al., 2019, who studied the triad wave-wave non-linear interactions in the case of long records of JONSWAP irregular waves (1200 $T_p$, where $T_P$ is the peak period) using a Fourier-based bispectral analysis. They found that the phase coupling is strong near the end of the slope, where second and third harmonics

become more important. They also noticed the appearance of low-frequency waves generated by the difference interactions during wave propagation. We must note here that the main difference between Fourier-based bicoherence and wavelet-based bicoherence is the number of degrees of freedom (Dong et al., 2008). Wavelet-based bicoherence is a suitable tool to detect

non-linear wave-wave interactions occurring in relatively short data sequences, and can be used to analyse data collected in laboratory flumes (Elsayed, 2006).

Most of the aforementioned studies were conducted in random wave conditions based on JONSWAP spectra. To the author's knowledge, few studies have attempted to quantify the degree of phase coupling resulting from the propagation of realistic spectrum wave trains in the nearshore zone using wavelet-based bicoherence. Experiments are performed on numerous Pierson-Moskowitz and JONSWAP wave trains propagating from a constant intermediate water depth to shoaling and breaking zones.

The paper is outlined as follows. The experimental set-up and test conditions are illustrated in section 2. In section 3, a short formal description of wavelet analysis and wavelet-based bicoherence is provided. The spatial evolution of wavelet-based bicoherence is discussed in section 4. Section 5 is devoted to conclusions and perspectives.

## 2 Experimental setup and wave train parameters

The following is a brief consideration of present wave trains generation; more details of the experiments can be found in
Abroug et al., 2020. The experiments were conducted in a two-dimensional wave flume of the M2C (Morphodynamique Continentale et côtière) Laboratory at Caen University, France. The flume is 22 m long, 0.8 wide and the water depth is $h_0 = 0.3$ m (Fig. 1). In this study, the relative water depth $k_p h_0 < 1.363$ is verified, which means that the modulation instability effect can be neglected (Janssen et al., 2007; Fedele et al., 2019). An Edinburgh Designs Ltd piston type wavemaker is located at one end of the flume to implement wave trains using linear wave generation signal. Wave trains are generated with almost
no reflection at the end of the flume, since measurements are performed before reflected waves travel back to the measurement location. Thus, the occurrence of resonant interactions potentially driven by reflected waves is limited and we only focus on bound waves.

The data used in this work are issued from Abroug et al., 2020. The present study relates to seven wave train simulations based on the averaged JONSWAP spectra (i.e. with peak factor $\gamma = 3.3$ or 7) or Pierson-Moskowitz spectra with varied peak wave
periods $f_p$ and wave steepnesses $S_0$ (i.e. non-linearity). The linear NewWave theory (Tromans et al., 1991), which is able to generate targeted waves at a prescribed location and time by combining sinusoidal components of different frequencies, is used as input for the generated focused wave trains. This theory was validated at deep water locations, at intermediate water depth locations (Taylor and Williams, 2004) and at coastal regions (Whittaker et al., 2016); for $kh < 0.5$). In NewWave theory, the expected shape of a wave train is the autocorrelation function (Fourier Transform of the spectral density).
For each wave train, a large number of wave signals were recorded along the flume to accurately follow the wave evolution in space. The surface elevation is measured by two aligned wave gauges located from the longitudinal coordinate $x_{\min} = 4$ m to $x_{\max} = 14$ m, where $x = 0$ is defined as the mean position of the wavemaker. The positions of these wave gauges are clearly delineated in Fig. 1. The sampling rate is 50Hz and each record duration is 35 s with a sample interval of 0.02 s. The fast Fourier transform (FFT) was applied to each signal, resulting in 1750 frequency components over the range [0, $3f_p$] and with

a spectral resolution $\Delta f = 0.023$ Hz. The distance from the wave maker for the focusing point was set to 12 m from the wave

maker.

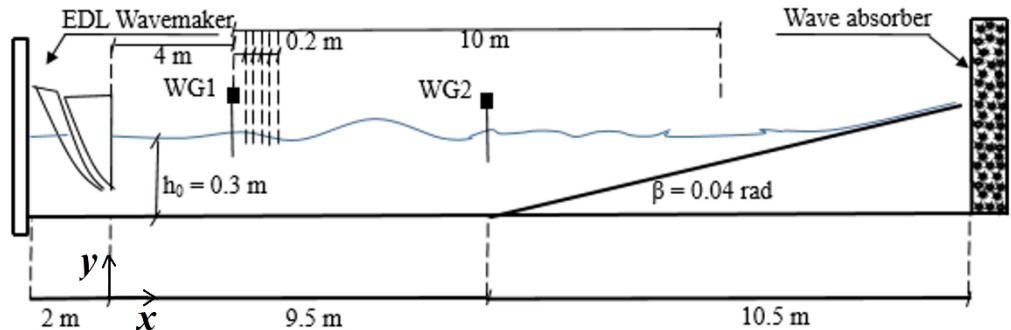

**Figure 1: Schematic experimental setup, WG1 and WG2 denote wave gauge n°1 and wave gauge n°2 respectively.**

Using linear NewWave theory, the free surface elevation of a wave train at a distance $x$ from the wavemaker can be written as

follows:

$$\eta(x,t) = \sum_{i=1}^{N} a_i \cos[k_i(x - x_0) - \omega_i(t - t_0)] \tag{1}$$

$$a_i = A_0 \frac{S(f_i)\Delta f}{\sum_{i=1}^{N} S(f_i)\Delta f} \tag{2}$$


where $a_i$ (Eq. (2)) is the amplitude of each component, $i$ varies from 1 to N (number of waves), $x_0$ and $t_0$ denote respectively

the predefined focal location and focal time, $k_i = \omega_i / g\tanh(k_i h)$ is the wavenumber, $\omega_i = 2\pi f_i$ is the angular frequency, $h$ is the

water depth, $A_0$ represents the theoretical linear crest amplitude of the wave train, $S(f_i)$ is the spectral density and

$\Delta f = \frac{f_{max} - f_{min}}{N-1}$ is the frequency step. JONSWAP and Pierson-Moskowitz are the two used spectra to represent the sea state.

All generated waves are crested focused waves, i.e. the phase angle of the wave group within its envelope at the focus position

is equal to zero.

Based on Eq. (1), the varied parameters during these experiments were the spectrum type ($S(f_i)$) and the wave steepness $S_0$.

The peak frequency parameter was chosen in order to have a relative depth $k_p h_0$ varying between 0.79 and 0.92 (Deep side in

Table 1). Deep and shallow sides in Table 1 represent respectively the flat bottom depth (4 m $< x <$ 9.5 m) and the shallowest

studied depth ($x = 14$ m). Five of the studied wave trains have more than one breaking and breaking locations $x_b$ are indicated

as bracketed intervals in Table 1.


**Table 1. Wave trains key parameters**

| Test | $f_p$ (Hz) | $S_0$ | Spectrum type | $x_b$ (m) | $k_p h_0$ Deep Side | $k_p h$ Shallow Side |
|------|-----------|-------|---------------|-----------|---------|---------|
| 1 | 0.70 | 0.19 | Gaussian | [11.85;12.55] | 0.84 | 0.34 |
| 2 | 0.66 | 0.14 | Pierson-Moskowitz | 12.9 | 0.79 | 0.31 |
| 3 | 0.66 | 0.28 | Pierson-Moskowitz | [11.09;11.82] | 0.79 | 0.31 |
| 4 | 0.75 | 0.25 | JONSWAP ($\gamma$=3.3) | [12.13;12.81] | 0.92 | 0.37 |
| 5 | 0.75 | 0.38 | JONSWAP ($\gamma$=3.3) | [10.5;11.61] | 0.92 | 0.37 |
| 6 | 0.75 | 0.11 | JONSWAP ($\gamma$=7) | 13.5 | 0.92 | 0.37 |
| 7 | 0.75 | 0.23 | JONSWAP ($\gamma$=7) | [12.07;12.69] | 0.92 | 0.37 |

## 3 Wavelet-based analysis

The free surface elevation of each wave train was studied through the bispectral analysis applied on short time series, via the

wavelet-based bicoherence. The detailed characteristics of the wavelet-based bicoherence can be found in Van Milligen et al., 1995 and a brief introduction of this technique is given below. The continuous wavelet transform $WT$ (a,τ) of a time series $f(t)$ is calculated as:

$$WT(a,\tau) = \int_{-\infty}^{+\infty} f(t)\psi_{a,\tau}^* \, dt \tag{3}$$

$$\psi_{a,\tau}(t) = |a|^{-0.5}\psi(\frac{t-\tau}{a}) \tag{4}$$

where the asterisk denotes the complex conjugate and $\psi_{a,\tau}$ (Eq. (4)) represents the mother wavelet function dilated by a factor $\tau$ and scaled by a factor $a$, $a > 0$. The latter parameter can be interpreted as the frequency inverse, i.e. $f = 1/a$. The wavelet transform can be interpreted as a series of bandpass filter of the time series with a mother wavelet function. We have chosen

the Morlet wavelet as a mother wavelet function because it provides information about phase and amplitude, and it is adapted for capturing coherence between harmonic components. The Morlet wavelet can be considered as a modulated Gaussian waveform and is defined as:

$$\psi(t) = \pi^{-1/4}e^{-\frac{t^2}{2}}e^{(i\omega_0 t)} \tag{5}$$


where $\omega_0$ denotes the dimensionless frequency and $t$ is the dimensionless time. The Morlet wavelet with $\omega_0 = 6$ is a good choice, since it ensures a good balance between time and frequency localisation (Grinsted et al., 2004; Dong et al., 2008). For the Morlet wavelet the scale $a$ is almost equal to the Fourier period $T$ ($T = 1.03\ a$). As mentioned in Dong et al., 2008, it is convenient to write the scales $a$ as fractional powers of two (Torrence and Compo, 1998):


$$a_i = a_0 2^{i\delta}, i = 0,1,2, \dots, M \tag{6}$$

$$M = \frac{1}{\delta} log_2(\frac{N\Delta t}{a_0}) \tag{7}$$

where $a_0$ is the smallest resolvable scale, M represents the largest scale and $\delta$ denotes the scale factor. The $a_0$ parameter should be chosen equal to $2 \times \Delta t$ (Torrence and compo, 1998; Dong et al., 2008). $N$ and $\Delta t$ represent respectively the number of points in the times series and the time sampling. The scale factor $\delta$ should be sufficiently small to provide high resolution and adequate sampling in scale. Moreover, for the Morlet wavelet, a scale factor $\delta = 0.5$ is the largest value that gives adequate sampling (Dong et al., 2008). It is for that reason that we opted for a scale factor $\delta = 0.02$, giving a total of 395 scales ranging from 0.04

up to 11.83 for respectively high and low frequency. The wavelet-based bispectrum (Eq. (8)) measures the phase coupling in the interval $\Delta T = 35$ s that occurs between $f_1$, $f_2$ and $f_3$ where the latter parameters must satisfy the frequency sum rule (Eq. (9)). Quadratic non-linear coupling occurs between $f_1$ and $f_2$, generating a third component at the sum frequency $f_3$.

The bispectrum (Eq. (8)), which is the double Fourier transform of the third-order moment, measures the extent of phase coherence due to the non-linear triad interaction between three waves that satisfy the frequency and phases matching criteria

(Eq. (9) and (10)). The estimation of wavelet-based bispectrum in the whole bifrequency plan can be based on its values in the interval $\psi : \{f_1 > f_2 > 0, f_1 + f_2 = f_s = 25$ Hz (Nyquist sampling frequency)$\}$.

$$B(a_1, a_2) = \int WT_x(f_1, \tau)\, WT_x(f_2, \tau) WT_x^*(f_3, \tau) d\tau \tag{8}$$

$$f_3 = f_1 + f_2 \tag{9}$$
$$\varphi_3 = \varphi_1 + \varphi_2 \tag{10}$$

The wavelet-based bicoherence (Eq. (11)), which can be defined as the normalised wavelet bispectrum, is used in practise to measure the degree of phase coupling (Larsen et al., 2001) and is bounded by 0 and 1 by the Schwarz inequality. A value close

to unity reveals a maximum amount of coupling and a value close to zero corresponds to a random phase relation.

$$b^2(a_1, a_2) = \frac{|B(a_1,a_2)|^2}{\left[\int_{t=0}^{t=35}|WT_x(a_1,\tau)WT_x(a_2,\tau)|^2 d\tau\right]\int_{t=0}^{t=35}|WT_x(a_3,\tau)|^2 d\tau} \tag{11}$$

Figure 2 exhibits a simple illustration of the wavelet-based bicoherence of a narrow-banded Gaussian wave train (Test 1)
recorded at $x = 4$ m from the wavemaker. The shading indicates the strength of non-linear coupling; dark red ($b^2(f_1, f_2) = 1$)
being totally coupled and dark blue ($b^2(f_1, f_2) = 0$) completely uncoupled. The degree of phase coupling is represented by the
colorbar indicating the sum interactions between two frequencies. In this manner, a visualisation of the non-linear activity
across the wave train propagation is feasible, detecting the frequency sections of the signal that contribute the most to the non-
linear activity. The two frequencies $f_1$ and $f_2$ are normalised by the peak frequency $f_p$. Red ($b^2(f_p, f_p)$) and yellow peaks represent
the phase coupling of the primary frequency component with its harmonic. In general, a non-null bicoherence $b^2(f_1,f_2) > 0$
means that the $f_3 = f_1+f_2$ component gains energy from the $f_1$ and $f_2$ components.

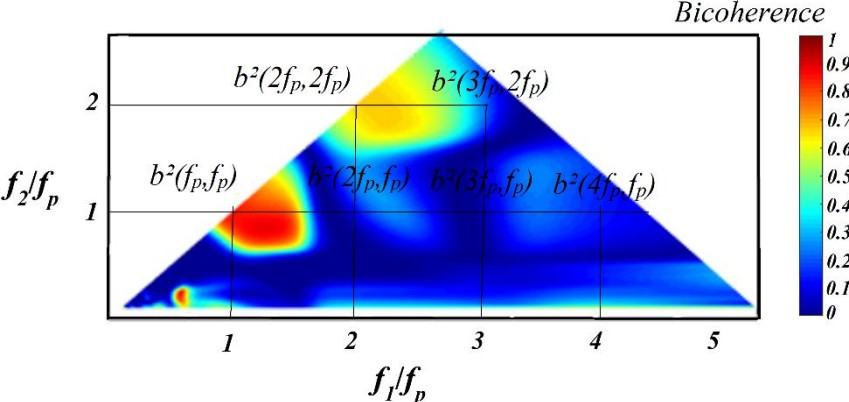

Figure 2: The wavelet-based bicoherence of a narrow-banded Gaussian wave train (Test 1) at x = 4 m.

## 4 Results and discussions

Figure 3 shows three sets of time series of three wave trains with approximately the same steepness $S_0$ and derived from
Pierson-Moskowitz (Test 3; $x_b \in$ [11.09; 11.82]), JONSWAP ($\gamma = 3.3$) (Test 4; $x_b \in$ [12.13; 12.81]) and JONSWAP ($\gamma = 7$)
(Test 7; $x_b \in$ [12.07; 12.69]) spectra at eight different locations along the flume. This preliminary figure shows surface elevation
time histories including the first measurement ($x = 4$ m), the propagation along the flat bottom, the shoaling and the breaking
of the focused wave group. It should be noted here that the seven studied wave trains are crest-focused wave groups ($\Phi = 0$).
Figure 4 shows, in a log scale, the spatial evolution of the Fourier spectra of the same three wave trains (Tests 3, 4 and 7). A
spatial downshift of the spectral peak (Cases 4 and 7), a steepening of the low frequency side and a widening of the high
frequency side are illustrated. These spectral variations, identified and quantified in Abroug et al., 2020, concern high and low
frequency components. The shift of energy is essentially due to non-linear wave-wave interactions among wave frequency
components during the focalisation process. Nevertheless, we do not distinguish which wave components participate in the
wave-wave interactions, nor do we distinguish the wave modes that undergo the strongest non-linear interactions.
Consequently, the wavelet-based bicoherence is used herein to provide information about the non-linear triad wave interactions
that cannot be easily obtained from the Fourier analysis which was used in Abroug et al., 2020.

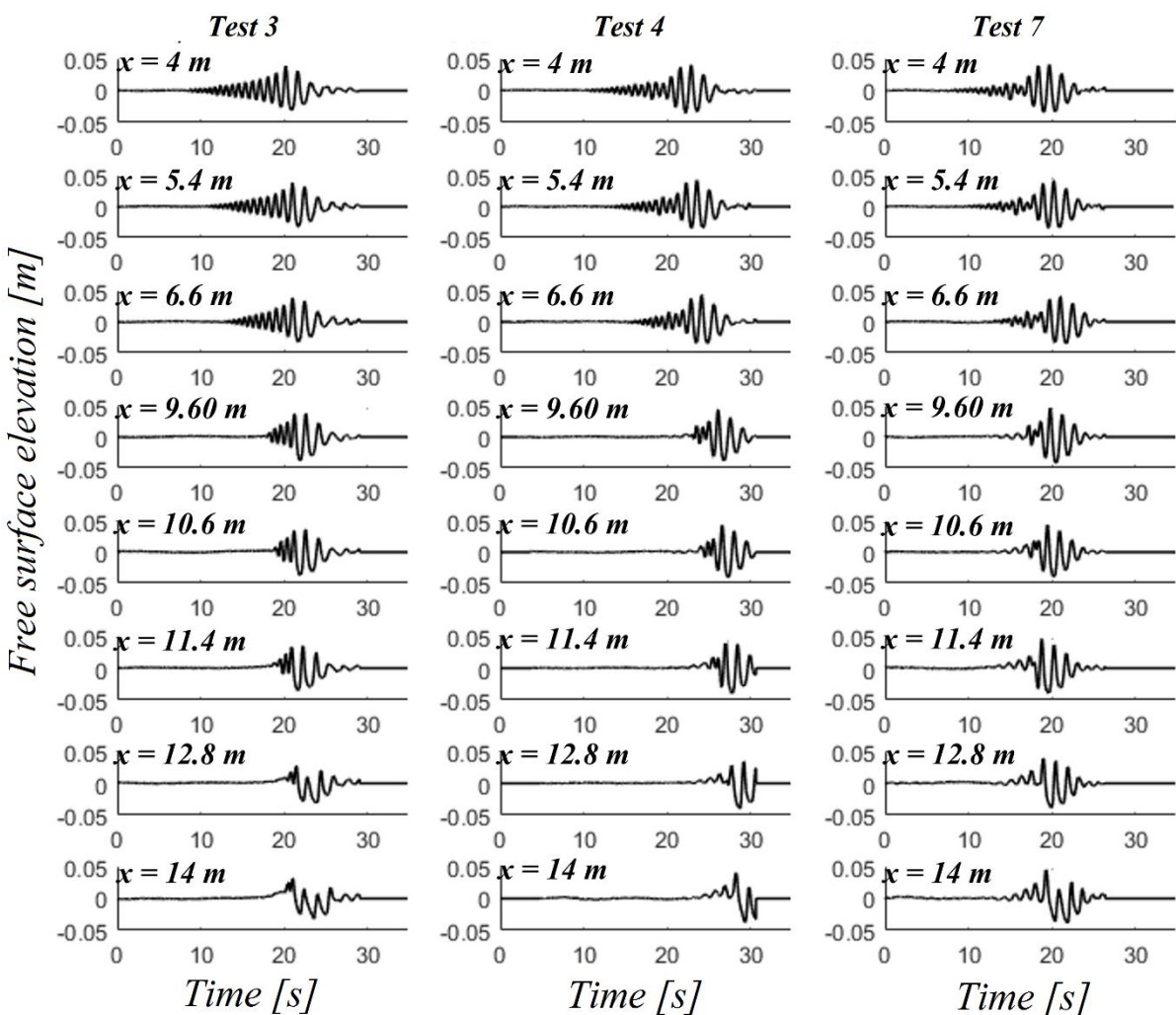

**Figure 3: Three sets of time series of Pierson-Moskowitz (Test 3), JONSWAP ($\gamma$ = 3.3) (Test 4) and JONSWAP ($\gamma$ = 7) (Test 7) wave trains.**

Figure 5 presents the spatial evolution of the wavelet-based bicoherence of a Pierson-Moskowitz wave train (Test 3; $x_b \in$ [11.09; 11.82]) along the flat bottom. This figure shows that wave-wave interactions between different modes are weak on flat bottom (4 m < x < 9.5 m; $k_p h_0$ = 0.79) and few frequency components participate in the focusing process. In the intermediate water depth region (4 m < x < 9.5 m), the sea state is almost Gaussian, and for that reason non-linear wave-wave interactions are relatively moderate. For example, $b^2(f_p, f_p)$ = 0.1 and $b^2$ $(f_p, 3f_p)$ = 0.065 at x = 4 m, indicate respectively a weak self-self wave interaction at the energy-frequency peak coupled with the energy at $2f_p$ and a very weak wave interaction at the peak frequency coupled with the energy at $4f_p$ (Fig. 5a). A significant bicoherence magnitude band ranging from $0.5f_p$ to $f_p$ is observed, i.e. $b^2(0.5f_p$ - $f_p$, $0.5f_p$ - $f_p)$, which indicates an energy transfer from low frequency components to the spectral peak. This partially explains the spatial evolution of the spectrum, namely the increase of energy in the peak region, which is

potentially a way of compensating the energy dissipation in the transfer region, i.e. the region between the spectral peak and high frequency regions, (Abroug et al., 2020; Liang et al., 2017). Note that the magnitude of the bicoherence is consistent with the fact that spectrum shape does not vary substantially along the flat bottom (4 m $< x <$ 9.5 m) (Fig. 4).

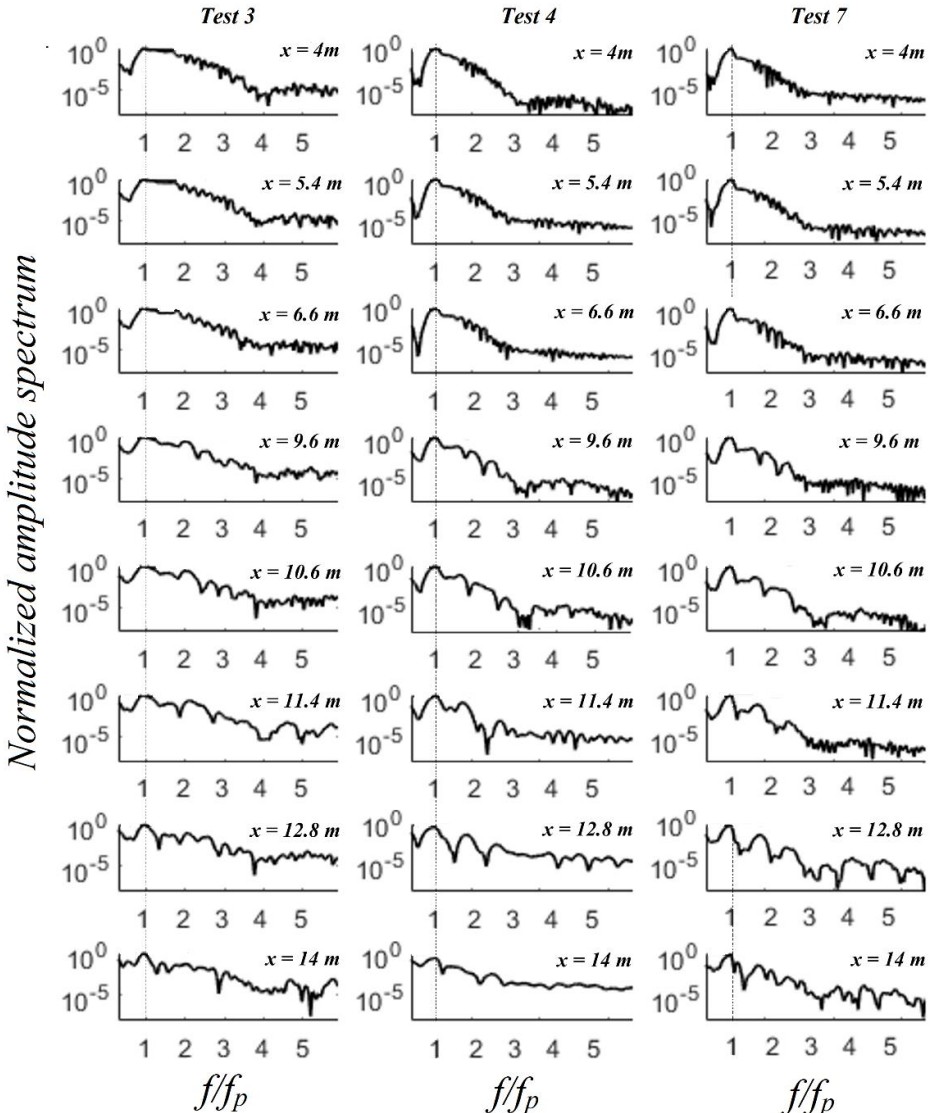

**Figure 4 : Spatial evolution of normalised amplitude spectra in a log scale for Tests 3, 4 and 7.**

As the wave train approaches the toe of the slope ($x \sim$ 9.5 m), more and more wave components are involved in the nonlinear phase coupling and the bicoherence values increase progressively. For $x =$ 9.6 m, just a little over the toe of the slope, the bicoherence magnitude among primary components increases slightly, i.e. $b^2(f_p, f_p) =$ 0.24 and $b^2(3f_p, f_p) =$ 0.15, which is consistent with the small energy increase in the high frequency region (Abroug et al., 2020).

As the wave train propagates in the shallower region (9.5 m $< x < x_b \in$ [11.09; 11.82]), the degree of phase coupling is seen to
increase rapidly (Fig. 6a). The degree of phase coupling within the peak frequency increases considerably at shallower regions
compared to deeper regions. Wave energy transfers increase in high frequency region and as a result, the spectrum broadens.
In the vicinity of the breaking location ($x_b \in$ [11.09; 11.82]), the non-linear coupling spreads over most of the wave
components. The increase of the second and third harmonic is clearly noticeable in Fig. 6b. The values of bicoherence for
approximately all frequency pairs are greater than 0.13, indicating that the non-linear coupling reaches its maximum level,
which means that almost all of the higher harmonics waves are involved in the propagation process.

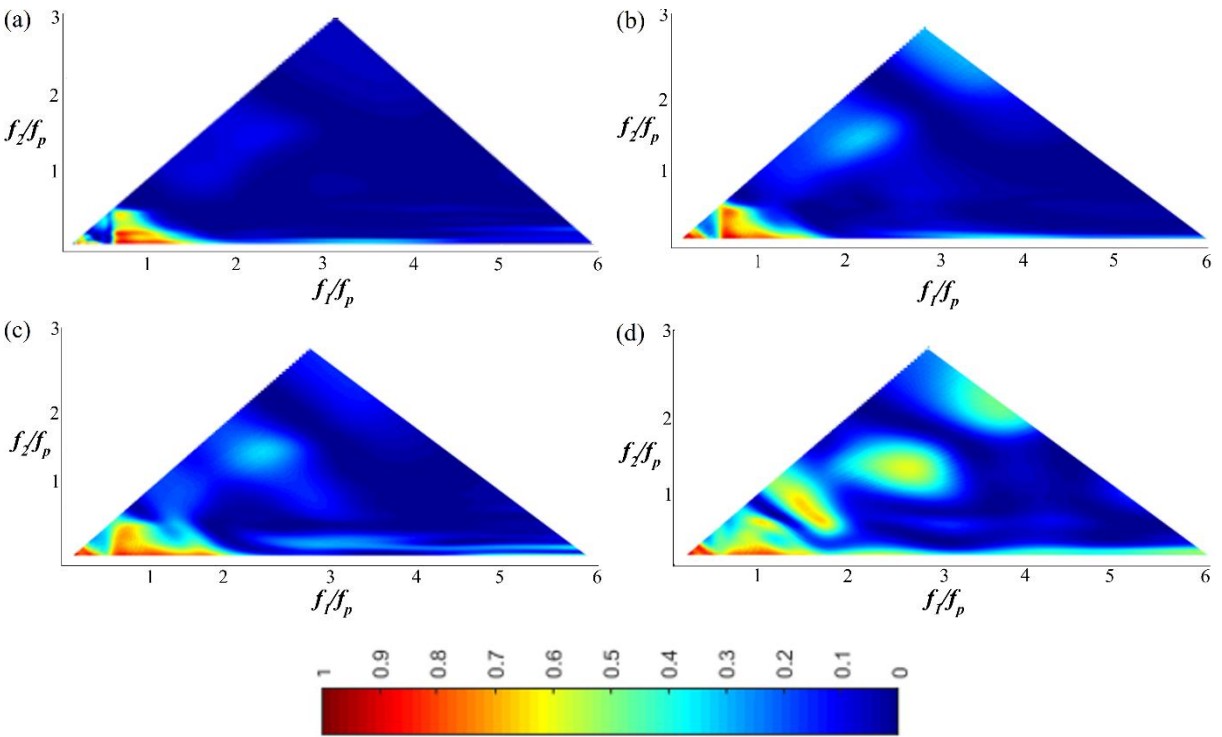

**Figure 5 : The wavelet-based bicoherence spatial evolution on the flat bottom for a Pierson-Moskowitz wave train (Test 3). (a):**
**$x = 4$ m; (b): $x = 5$ m; (c): $x = 8$ m; (d): $x = 9.6$ m.**

Downstream the breaking location ($x > x_b \in$ [11.09; 11.82]), the degree of phase coupling between frequency components
decreases drastically and the bicoherence becomes less structured (Fig. 6d). This result is consistent with the decreasing trend
of energy in higher frequency components downstream the breaking location (Tian et al., 2011; Abroug et al., 2020).

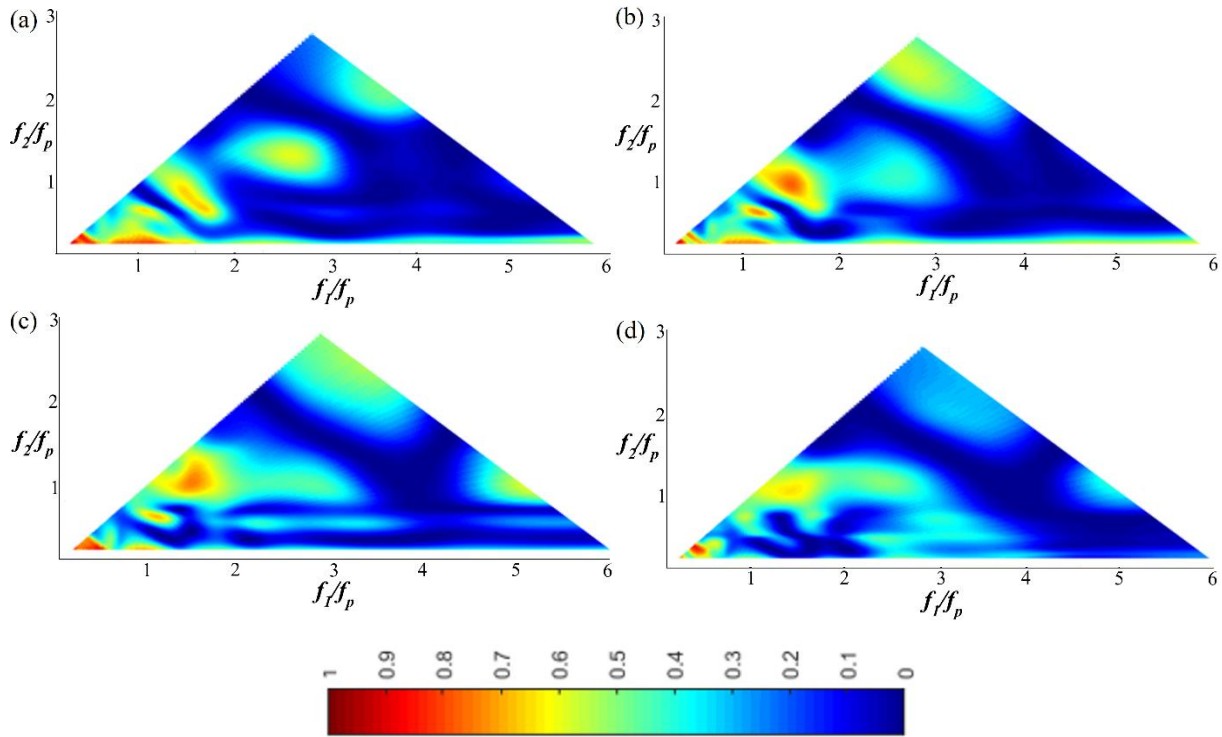

Figure 6 : The wavelet-based bicoherence spatial evolution on the sloping bottom for a Pierson-Moskowitz wave train (Test 3). (a): $x = 11$ m; (b): $x = 12$ m; (c): $x = 13$ m; (d): $x = 13.8$ m.

In fig. 7 and 8, a JONSWAP ($\gamma = 3.3$) wave train (Test 5; ; $x_b \in [10.5; 11.61]$) is chosen to illustrate the spatial evolution of the wavelet-based bicoherence of a narrower wave train propagating over the flat and the sloping bottom. Wave-wave interactions evolve qualitatively in the same way compared to the case of Pierson-Moskowitz. Fig. 7a ($x = 4$ m) shows that the two dominant phase coupling peaks appear at the bifrequencies ($f_p$, $f_p$) and ($0.5f_p$, $0 - 0.5f_p$), which illustrates that the quadratic non-linear interactions only occur between the peak and low-frequency modes. Note that no other peak was found to be significant. As the wave train propagates over the shallower region ($x > 9.5$ m), new phase couplings appear at the bifrequencies ($2f_p$, $f_p$), ($3f_p$, $f_p$) and ($2f_p$, $3f_p$) (Fig. 8). This finding illustrates that quadratic non-linear interactions between the peak frequency, the first harmonic, the second harmonic and third harmonics result from the gradual broadening of the spectrum. It is an accordance with previous studies demonstrating that energy is mainly transferred to high frequencies during the shoaling process (Tian et al., 2011; Liang et al., 2017; Abroug et al., 2020). For this wave train (Test 5), the wavelet-based bicoherence reaches its maximum shortly after the breaking ($x_b \in [10.5; 11.61]$) at $x = 12$ m. For example, b²($f_p$, $f_p$) = 0.7, b²($2f_p$, $f_p$) = 0.53 and b²($2f_p$, $2f_p$) = 0.16. Triad interactions lead to skewed wave profiles and can characterize the near-breaking conditions (Fig. 3 for $x > 10.6$ m).


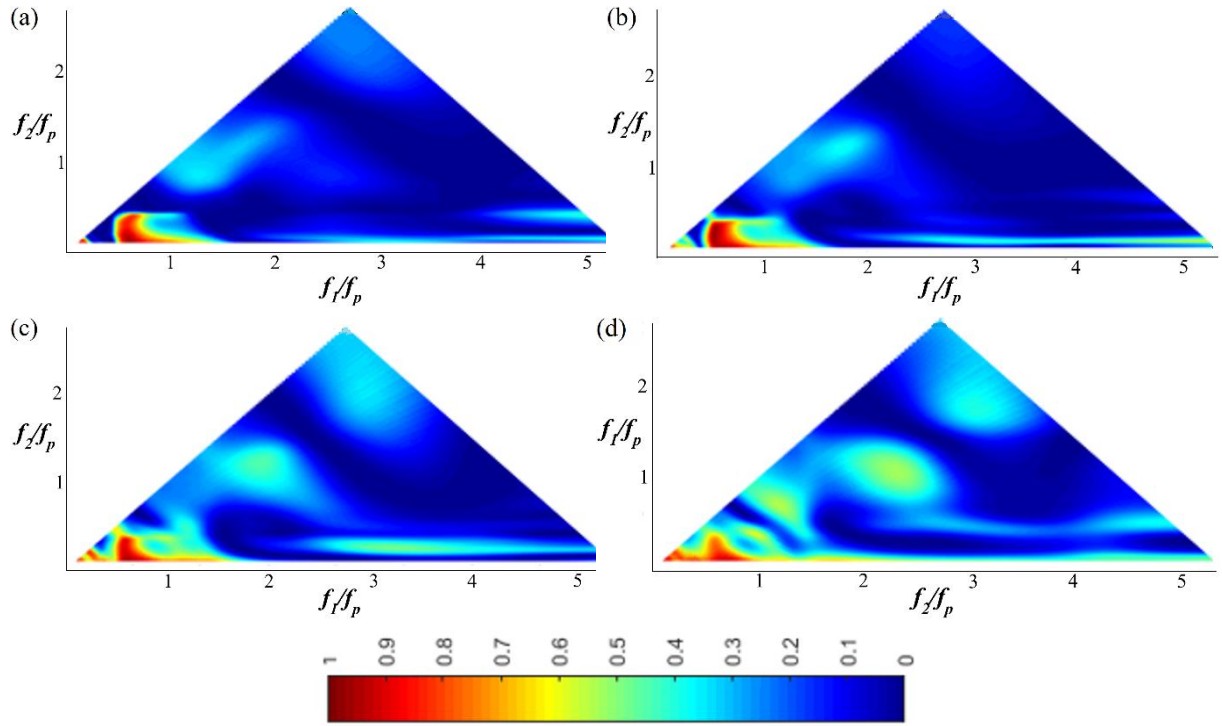

**Figure 7: The wavelet-based bicoherence spatial evolution on the flat bottom for JONSWAP ($\gamma = 3.3$) wave train (Test 5). (a): $x = 4$ m; (b): $x = 6$ m; (c): $x = 8$ m; (d): $x = 9.6$.**

Beyond the breaking location ($x > x_b \in [10.5;11.61]$), the bicoherence decreases sharply and becomes less structured. For
example $b^2(f_p, f_p) = 0.52$, $b^2(2f_p, f_p) = 0.31$ and $b^2(2f_p, 2f_p) = 0.004$ at $x = 13.6$ m, i.e. $h = 0.13$ m. This pattern is qualitatively
similar to that obtained in the case of a Pierson-Moskowitz wave train. This indicates that the increasing trend of the phase
coupling is one of the more important reasons for the wave train breaking in shallow water.

Figures 9 and 10 depict the wavelet-based bicoherence spectra for the case of a JONSWAP ($\gamma = 7$) wave train
(Test 7; $x_b \in [12.07; 12.69]$) at eight locations along the wave flume. No bispectral peak appears at $b^2(2f_p, f_p)$ and this is may
be not surprising as no clear third harmonic $3f_p$ is present in the frequency spectrum (Fig. 4). Furthermore, wavelet-based
bicoherence diagrams show that the phase coupling reaches its maximum level at frequencies slightly higher than the exact
harmonics ($2f_p$, $3f_p$ ...). This result is consistent with the results of Ma et al., 2010 who explained this process by the slight
upshift of peak values in spectrum at higher harmonics, which is readily seen in Fig. 4. The fact that clear 1[st], 2[nd] and 3[rd]
harmonics are not present, is possibly due to other mechanisms such as quadruplet interactions ($f_1+f_2 = f_3+f_4$, Elgar et al., 1995)
which have a shape stabilizing impact on the spectrum and are confined to free waves. This result is consistent with the peak
frequency downshift demonstrated experimentally in Stansberg, 1994 and Abroug et al., 2020, where it was interpreted as a
self-stabilizing feature.

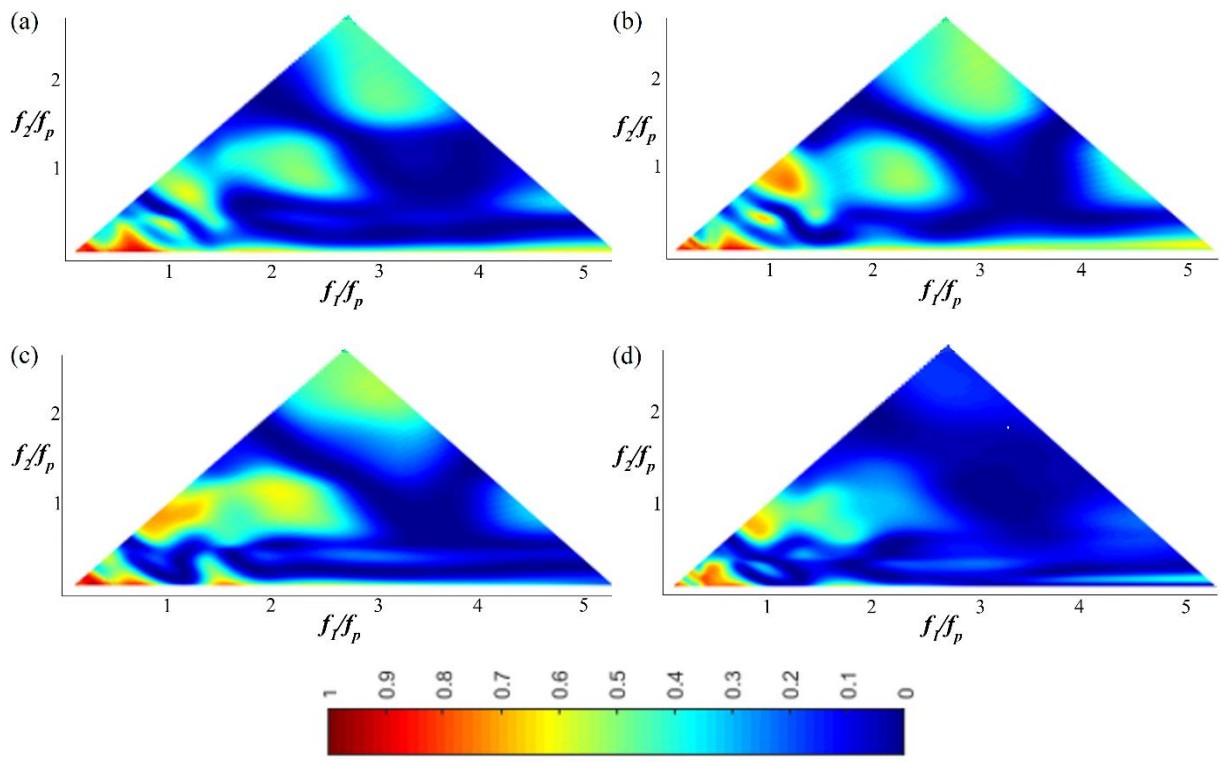

**Figure 8: The wavelet-based bicoherence spatial evolution on the sloping bottom for JONSWAP ($\gamma$ = 3.3) wave train (Test 5). (a):** $x$ **= 10.20 m; (b):** $x$ **= 11.20 m; (c):** $x$ **= 12.40 m; (d):** $x$ **= 13.60 m.**

Figure 11 summarizes the variability in the location and intensity of the wavelet-based bicoherence between the bifrequency
pairs $(f_p, f_p)$, $(2f_p, f_p)$, $(3f_p, f_p)$, $(4f_p, f_p)$, $(2f_p, 2f_p)$ for several tests. The two vertical solid lines and the dotted line respectively indicate the breaking region and the toe of the slope. This figure indicates that the steepness has a strong influence on the non-linear phase coupling between harmonics in intermediate water depth ($h_0$ = 0.3 m). Non-linear wave-wave interactions and their increasing trend is more important for wave trains having strong non-linearities. Beyond the wave breaking ($x > x_b$), the decreasing trend of the phase coupling between harmonics is also more significant in the case of strong steepness $S_0$. This
result is in accordance with the dissipation related to breaking, which is particularly noticeable when the wave steepness is high (Abroug et al., 2020).

An important similarity between different spectra, is that important wave-wave interactions are mostly limited to the first harmonics of primary waves $(f_p, f_p)$, $(2f_p, f_p)$. This finding is consistent with the energetic behavior of wave trains downstream the wave breaking (Abroug et al., 2020). Moreover, in the case of small and moderate wave steepness (Test 2; $x_b$ = 12.9 and
Test 6; $x_b$ = 13.5), the phase coupling varies slightly downstream the wave breaking compared to that found prior the breaking, suggesting that a small energy transfer happens downstream the breaking location.

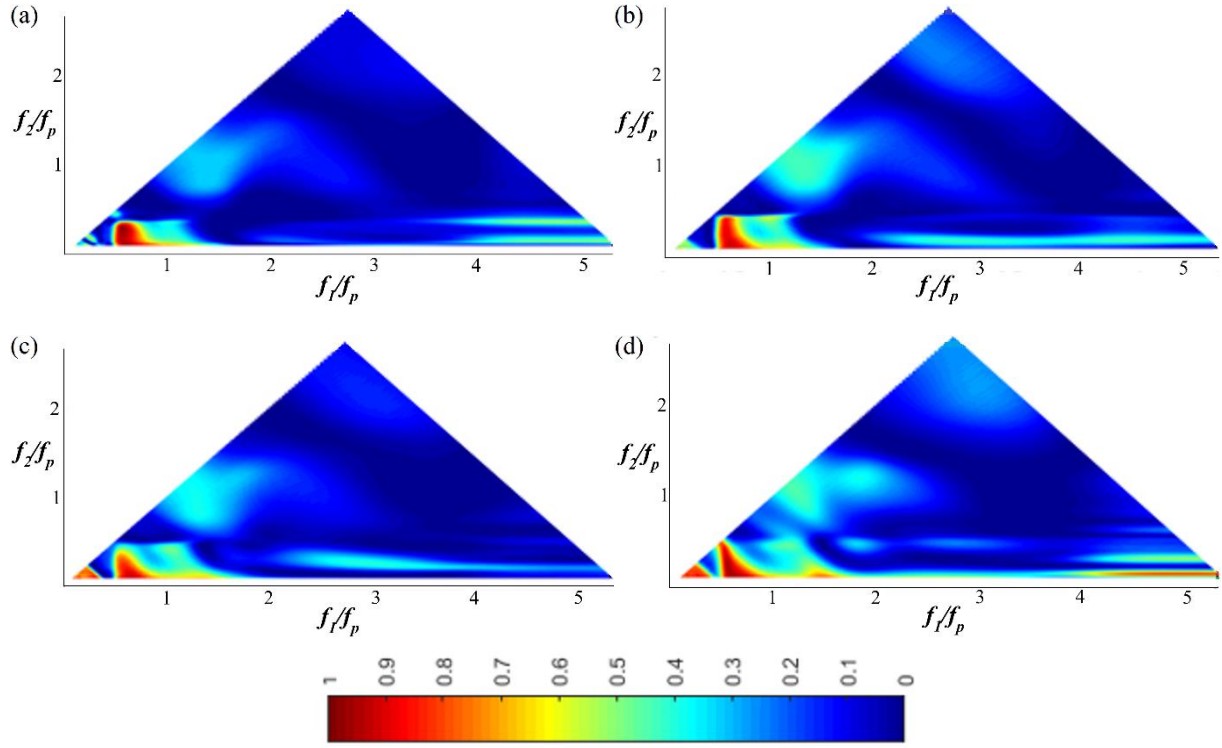

**Figure 9: The wavelet-based bicoherence spatial evolution on the flat bottom for JONSWAP ($\gamma = 7$) wave train (Test 7). (a): $x = 4$ m; (b): $x = 6.4$ m; (c): $x = 8$ m; (d): $x = 9.6$ m.**

It can be concluded that bound or non-resonant interactions play an important role in the evolution and breaking of wave trains in shallow water depth. Although the bound waves are not supposed to contribute to the energy redistribution, our experimental observations raise the question of the impact of bound interactions on dissipation and energy transfers among different

frequency components.

## 5 Conclusions and perspectives

An experimental approach is proposed for determining the non-linear wave-wave interactions, which accompany the propagation of large amplitude wave trains, that might cause damage to coastal zones, marine structures and navigation vessels. We investigate seven focused wave trains derived from JONSWAP ($\gamma = 3.3$ or 7) and Pierson-Moskowitz spectra propagating

from intermediate water depth to the inner surf zone. The results presented in this study extend the parameter range of observations of triad interactions. The experimental conditions were selected based on two parameters: the wave steepness and the spectrum type. The present data were collected in intermediate water with a $k_p h_0$ varying between 0.92 and 0.79. A typical wave train consists of a large number of waves interacting with one another. Wavelet-based bicoherence is used to investigate

the phase coupling between frequency components of short time series. Some consequences of non-linear transfer are briefly discussed; in particular the role played by non-linear interactions in shaping the high frequency part of the spectrum, the relative contribution of each harmonic and the downshifting of the peak spectrum demonstrated in previous studies. Note that our experimental study is different from previous experiments (Dong et al., 2008; Ma et al., 2010) regarding the slope geometry and most importantly, the use of three different spectral types.

Along the flat bottom (4 m < $x$ < 9.5 m), one might assume that the influence of triad interactions is very weak for the three considered spectra. The bispectral analysis of the data shows that as the waves propagate along the flat bottom, the magnitude of the bicoherence increases slightly (between 0% and 20% of its initial value). Moreover, this is foreseeable because the spectrum and the wave train shape do not substantially change along the flat bottom and a small amount of energy is transferred from the peak region to high frequency components.

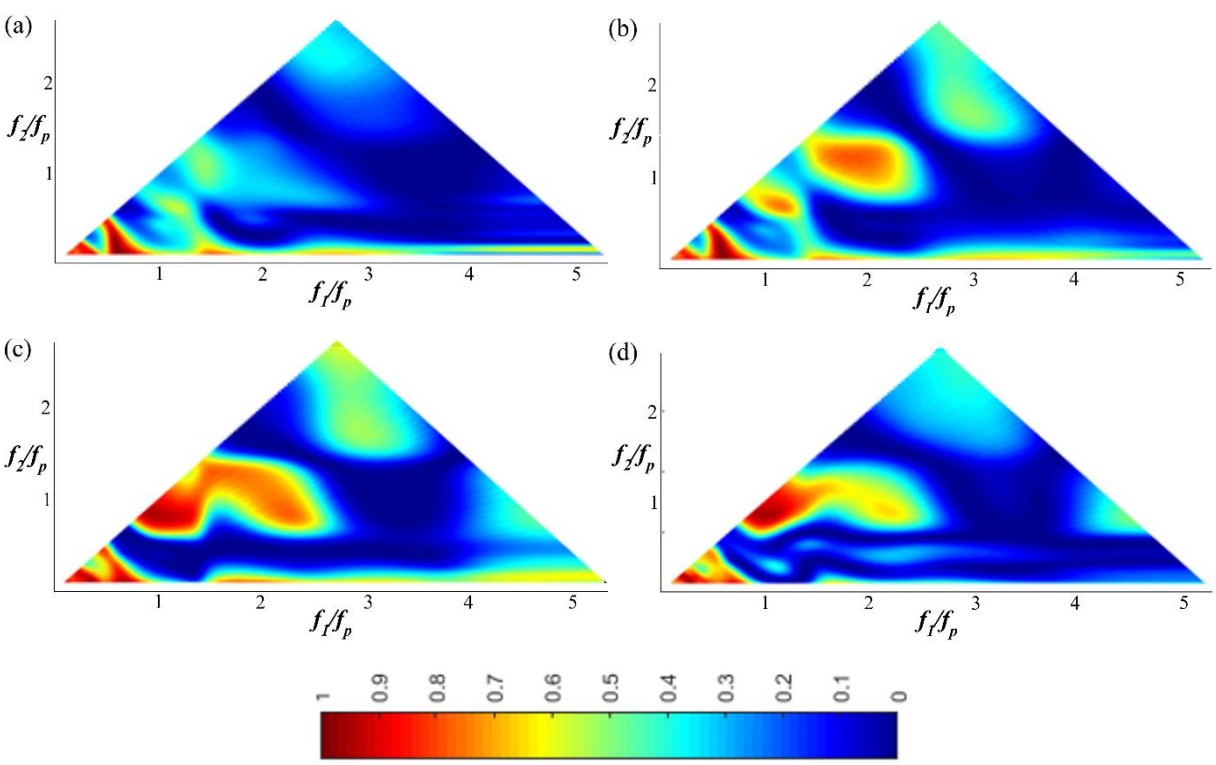

Figure 10: The wavelet-based bicoherence spatial evolution on the sloping bottom for JONSWAP ($\gamma$ = 7) wave train (Test 7). (a): $x$ = 10.2 m; (b): $x$ = 11 m; (c): $x$ = 12.4 m; (d): $x$ = 13.8 m.

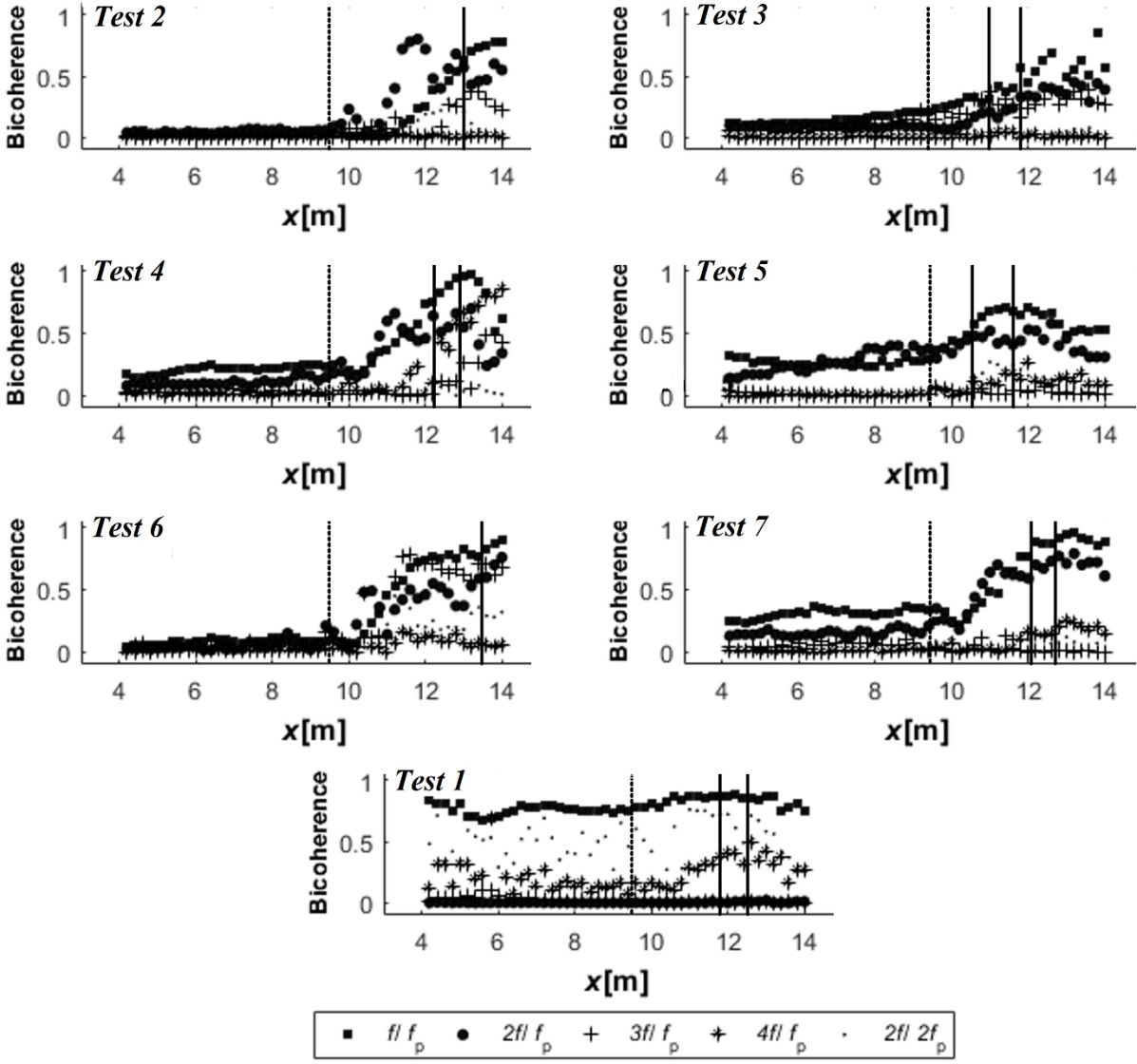


**Figure 11: Spatial variation of the wavelet-based bicoherence among harmonics. The two vertical solid lines and the dotted line respectively indicate the breaking region and the toe of the slope.**

When the wave train reaches the slope ($9.5\ m < x < x_b$), wave-wave interactions among high order harmonics increase rapidly and reach the maximum degree in the breaking/focus location. In line with previous studies (Elsayed, 2006; Dong et al., 2008;

Ma et al., 2010), strong nonlinear interactions were predominantly observed in the shallower region. The analysis showed a gradual broadening of the bicoherence spectrum, which is in accordance with previous studies who demonstrated that the energy is transferred mainly to high frequencies regions (Tian et al., 2011; Abroug et al., 2020). This is partly due to significant spectral transformations which are more important during the shoaling process. Particularly, this analysis showed a considerable contribution of 2nd and 3rd harmonics for unidirectional steep wave trains and the spectral components at the

second harmonic $2f_p$ have increased substantially (6 times its initial value). The bispectral analysis results show that the wave non-linearity $S_0$ plays an important role in the increasing trend of phase coupling, which is more important for wave trains having strong non-linearities. This last finding agrees well with the conclusions made by Ma et al., 2010.

An innovative aspect of this paper is presenting wavelet-based bispectral analysis for highly non-linear intermediate water waves with different spectral types. If we compare the three spectra, we can see that all nonlinear interactions on the flat bottom ($x < 9.5$ m) are weak ($b^2 < 0.15$) in the case of wide spectrum wave trains (Tests 2 and 3 Fig. 11). However, in the case of narrower spectra, more frequencies (e.g. $f_p$, $2f_p$ and $3f_p$) are implicated in the focusing process (Tests 4, 5, 6 and 7 Fig. 11) and the corresponding phase coupling is higher ($b^2 > 0.2$). This finding is in agreement with the stable behavior of wide spectrum wave trains, which was demonstrated experimentally in Abroug et al., 2019 and Stansberg, 1994. In intermediate water depth ($0.79 < k_p h < 0.92$), wide spectrum harmonics ($f_p$, $2f_p$, $3f_p$ …) are less implicated in the focusing process compared to narrow spectrum harmonics. In shallow water regions ($9.5$ m $< x < x_b$) and after breaking ($x_b < x$), the spatial evolution of the phase coupling is qualitatively similar for the three spectra.

The results obtained in this study show important features in wave-wave interactions during the propagation of focused waves. This study strengthens the usefulness of wavelet-based analysis in detecting features that are hidden in a Fourier-based analysis, and in explaining a number of phenomena, such as the process leading to wave breaking and the energy transfer between wave components. Nevertheless, in order to confirm the use of wavelet-based bicoherence for more realistic 3D studies with structures, efforts should be made to expand this study for example by investigating greater water depths, higher steepness and wider spectra. Furthermore, the observed evolution of bicoherence for focused waves should be compared to that of waves with similar steepness and bandwidth but with initial random distribution of phase. In other words, efforts should be made to identify and quantify the phase coupling differences between focusing wave trains and non-focusing waves. Information concerning the phase coherence can be obtained by calculating the biphase parameter ($\beta$ (a1, a2), Ma et al., 2010). It will be interesting to quantitatively measure the deviation of biphase values between primary waves/higher harmonics and to analyse their spatial evolution through different spectra to distinguish differences. Finally, a detailed study of how bound energy at harmonics would be influenced by quadruplet interactions should be performed.

Shallow water extreme waves are a major threat to offshore structures and ships. Findings in this study would improve our understanding of the propagation and breaking of extreme wave trains and help engineers in monitoring the wave propagation in coastal regions. The experimentally measured wave signals are highly nonlinear, unsteady and nonstationary. Consequently, the application of time-localized bicoherence analysis is shown to be a powerful approach. This study shows that an extreme wave can be readily identified from the wavelet-based bicoherence spectrum, in which strong energy is transferred to high-frequency components during the shoaling process. Such a detailed examination of individual nonlinear interactions is useful for practical applications such as investigating nonlinear responses of high frequency loads observed in severe sea conditions (e.g. springing and ringing which are excited by the sum-frequency components of irregular waves). By identifying which wave components are the most involved in the propagation process, this study may provide a complementary approach to existing experimental and field studies for determining extreme wave group runup and overtopping.

## Competing interests

The authors declare that they have no conflict of interest.

## Acknowledgments

This work has been carried out within the framework of the DYNAT (Dynamique du littoral et risques NATurels) project and has received funding from the Normandy region. We thank L. Perez and D. Mouazé for providing technical supports during the experiments. The authors wish also to express their gratitude to S. Baatout for her thorough re-reading of this article.

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
