# Peer review of "Laboratory study of non-linear wave-wave interactions of extreme focused waves in the nearshore zone"

_Natural Hazards and Earth System Sciences, 2020_

## Referee Comment (RC1) · Efim Pelinovsky (Referee) · 7 Aug 2020

The paper under review is devoted to the experimental study of nonlinear interactions in the focused wave packet by using the wavelet bi-spectral analysis. This experiment continues a series of studies done by these authors with the investigation the nonlinear-dispersive mechanism of the rogue wave formation. In their experiments the authors used the wave trains with various spectra (Pierson-Moskowitz and JONSWAP) in intermediate depth. Strong nonlinear effects are observed on the sloping beach as it is expected. Nonlinear energy transfer in the high-frequency region is analyzed. In fact, such processes have been actively studied earlier, and, perhaps, the novel moment

here is the demonstration of nonlinear effects through the bi-spectral analysis. To add to this, I would like the authors to formulate the obtained results in Conclusion better underlying their difference from the known results.

Being mostly a theoretician, I have two comments.

1. Equations (2) and (3) are written inaccurately. Function (3) does not contain tau and the parameter a.

2. For the wave focusing, it is necessary to vary the local frequency on the specific law for intermediate depth. I do not understand which formula for the local frequency versus time has been used. Perhaps, by using the optimal law, the focusing can occur on the flat bottom. If there is no specific focusing, there is an interference. Moreover, it should be reflected in the title.

To conclude, I may recommend the paper with the suggested revision.
* * *

---

## Referee Comment (RC2) · Yuxiang Ma (Referee) · 8 Aug 2020

This paper present a nonlinear phase coupling analysis of focusing wave groups propagating over a slope. Foucsing wave groups with three different spectra , Gaussian, P-M, JONSWAP were generated in a wave flume. But, only three wave groups are discussed. Acatually, the topic of this study is not new. Spectral and bi-spectral analysis of irregular waves over shoal have been presented previously. I think the new aspect of this paper is presenting bispectral analysis for waves with different spectral types. My comments are outlined as follow.

1. Page 2, lines 47-47. The authors argues that "It is important to mention here that

resonant interactions are not easily achieved in unidirectional wave train propagation since the resonant conditions cannot not be satisfied in a small area. ", I think it is better to specfiy that this is only true in shallow water. As for deep or intermediate water depth, resonant interactions is very strong in unidirectinal wave trains.

2. The reference Guohai et al. 2008 should be Dong et al. (2008).

3. Fig. 3, There is a mistake in captain: JONSWAP (gamma = 7) (Test 7)

4. Where is the incipient breaking points and spatial range should be presented in the text.

5. The authors should concentrate on the analysis of difference biphase coupling for different spectra?

---

## Author Comment (AC1) · 8 Sep 2020

The authors would like to thank the reviewer for his thoughtful and useful comments on our paper. We have considered all the suggestions. Our point-by-point responses (R) to comments and questions (Q) are detailed below:

Q1:

The paper under review is devoted to the experimental study of nonlinear interactions in the focused wave packet by using the wavelet bi-spectral analysis. This experiment continues a series of studies done by these authors with the investigation the nonlinear

dispersive mechanism of the rogue wave formation. In their experiments, the authors used the wave trains with various spectra (Pierson-Moskowitz and JONSWAP) in intermediate depth. Strong nonlinear effects are observed on the sloping beach as it is expected. Nonlinear energy transfer in the high-frequency region is analyzed. In fact, such processes have been actively studied earlier, and, perhaps, the novel moment here is the demonstration of nonlinear effects through the bi-spectral analysis. To add to this, I would like the authors to formulate the obtained results in Conclusion better underlying their difference from the known results.

R1:

We would like to thank the reviewer for his remark. We tried to formulate and synthesise the obtained results in order to distinguish similarities and differences from the known results. Here is the conclusion. Redundant sentences have been deleted as well.

An experimental approach is proposed for determining the non-linear wave-wave interactions, which accompany the propagation of large amplitude wave trains, that might cause damage to coastal zones, marine structures and navigation vessels. We investigate seven focused wave trains derived from JONSWAP ($\gamma$ = 3.3 or 7) and Pierson-Moskowitz spectra propagating from intermediate water depth to the inner surf zone. The results presented in this study extend the parameter range of observations of triad interactions. The experimental conditions were selected based on two parameters: the wave steepness and the spectrum type. The present data were collected in intermediate water with a kph0 varying between 0.92 and 0.79. A typical wave train consists of a large number of waves interacting with one another. Wavelet-based bicoherence is used to investigate the phase coupling between frequency components of short time series. Some consequences of non-linear transfer are briefly discussed; in particular the role played by non-linear interactions in shaping the high frequency part of the spectrum, the relative contribution of each harmonic and the downshifting of the peak spectrum demonstrated in previous studies. Note that our experimental study is different from previous experiments (Dong et al., 2008; Ma et al., 2010) regarding the slope

geometry and most importantly, the use of three different spectral types.

Along the flat bottom (4 m < x < 9.5 m), one might assume that the influence of triad interactions is very weak for the three considered spectra. The bispectral analysis of the data shows that as the waves propagate along the flat bottom, the magnitude of the bicoherence increases slightly (between 0

When the wave train reaches the slope (9.5 m < x < xb), wave-wave interactions among high order harmonics increase rapidly and reach the maximum degree in the breaking/focus location. In line with previous studies (Elsayed, 2006; Dong et al., 2008; Ma et al., 2010), strong nonlinear interactions were predominantly observed in the shallower region. The analysis showed a gradual broadening of the bicoherence spectrum, which is in accordance with previous studies who demonstrated that the energy is transferred mainly to high frequencies regions (Tian et al. 2011; Abroug et al., 2020). This is partly due to significant spectral transformations which are more important during the shoaling process. Particularly, this analysis showed a considerable contribution of 2nd and 3rd harmonics for unidirectional steep wave trains and the spectral components at the second harmonic 2fp have increased substantially (6 times its initial value). The bispectral analysis results show that the wave non-linearity S0 plays an important role in the increasing trend of phase coupling, which is more important for wave trains having strong non-linearities. This last finding agrees well with the conclusions made by Ma et al. (2010).

An innovative aspect of this paper is presenting wavelet-based bispectral analysis for highly non-linear intermediate water waves with different spectral types. If we compare the three spectra, we can see that all nonlinear interactions on the flat bottom (x < 9.5 m) are weak ($b^2 < 0.15$) in the case of wide spectrum wave trains (Tests 2 and 3 Fig. 11). However, in the case of narrower spectra, more frequencies (e.g. fp, 2fp and 3fp) are implicated in the focusing process (Tests 4, 5, 6 and 7 Fig. 11) and the corresponding phase coupling is higher ($b^2 > 0.2$). This finding is in agreement with the stable behavior of wide spectrum wave trains, which was demonstrated experimentally

in Abroug et al. (2019) and Stansberg (1994). In intermediate water depth (0.79 < kph < 0.92), wide spectrum harmonics (fp, 2fp, 3fp . . .) are less implicated in the focusing process compared to narrow spectrum harmonics. In shallow water regions (9.5 m < x < xb) and after breaking (xb < x), the spatial evolution of the phase coupling is qualitatively similar for the three spectra.

The results obtained in this study show important features in wave-wave interactions during the propagation of focused waves. This study strengthens the usefulness of wavelet-based analysis in detecting features that are hidden in a Fourier-based analysis, and in explaining a number of phenomena, such as the process leading to wave breaking and the energy transfer between wave components. Nevertheless, in order to confirm the use of wavelet-based bicoherence for more realistic 3D studies with structures, efforts should be made to expand this study for example by investigating greater water depths, higher steepness and wider spectra. Furthermore, the observed evolution of bicoherence for focused waves should be compared to that of waves with similar steepness and bandwidth but with initial random distribution of phase. In other words, efforts should be made to identify and quantify the phase coupling differences between focusing wave trains and non-focusing waves. Information concerning the phase coherence can be obtained by calculating the biphase parameter ($\beta$ (a1, a2), Ma et al., 2010). It would be interesting to quantitatively measure the deviation of biphase values between primary waves/higher harmonics and to analyse their spatial evolution through different spectra to distinguish differences. Finally, a detailed study of how bound energy at harmonics would be influenced by quadruplet interactions should be performed.

Q2: Equations (2) and (3) are written inaccurately. Function (3) does not contain tau and the parameter a.

R2:

We would like to thank the reviewer for this comment. I added Eq 4, which contains the

two parameters tau and a, in order to make Eq 3 more explicit (Line 130). Equation 4 has been added between the continuous wavelet transform WT (a,$\tau$) function and the Morlet wavelet function.

$\psi_{(a,\tau)}$ (t)=|a|$^{(}-0.5)\psi$((t-$\tau$)/a)

Q3:

For the wave focusing, it is necessary to vary the local frequency on the specific law for intermediate depth. I do not understand which formula for the local frequency versus time has been used. Perhaps, by using the optimal law, the focusing can occur on the flat bottom. If there is no specific focusing, there is an interference. Moreover, it should be reflected in the title.

R3:

It is an interesting comment and maybe the methodology of generation needs more clarifications from the authors. Here is the explanation, which was added to the manuscript.

Line (85-89)

The linear NewWave theory (Tromans et al. (1991)), which is able to generate targeted waves at a prescribed location and time by combining sinusoidal components of different frequencies, is used as input for the generated focused wave trains. This theory was validated at deep water locations, at intermediate water depth locations (Taylor and Williams (2004)) and at coastal regions (Whittaker et al. (2016); for kh < 0.5). In NewWave theory, the expected shape of a wave train is the autocorrelation function (Fourier Transform of the spectral density).

Line (100-111)

Using linear NewWave theory, the free surface elevation of a wave train at a distance x from the wavemaker can be written as follows:

$\eta$(x,t)=$\sum_{(}i=1_N a_i cos[k_i(x-x_0)-\omega_i(t-t_0)]$(1)$a_i = A_0(S(f_i)\Delta$f)/($\sum_{(}i=1_N S(f_i)\Delta$fãĂŮ) (2)

where ai (Eq. (2)) is the amplitude of each component, i varies from 1 to N (number of waves), x0 and t0 denote respectively the predefined focal location and focal time, ki = $\omega$i / gtanh(kih) is the wavenumber, $\omega$i = 2$\pi$fi is the angular frequency, h is the water depth, A0 represents the theoretical linear crest amplitude of the wave train, S(fi) is the spectral density and $\Delta$f = (f$_{max} - f_{min})/(N - 1) is the frequency step. The wave group is generated with a given linear focus position (x = 12m from the wavemaker) based on linear focusing in a constant water depth.$

So to answer your question, yes, by modifying x < 9.5 m in the EDL Software, we can obtain a focusing on the flat bottom.

JONSWAP and Pierson-Moskowitz are the two spectra used to represent the sea state. All generated waves are crested focused waves, i.e. the phase angle of the wave group within its envelope at the focus position is equal to zero.

Please also note the supplement to this comment:
https://nhess.copernicus.org/preprints/nhess-2020-209/nhess-2020-209-AC1-supplement.pdf

---

## Author Comment (AC2) · 8 Sep 2020

The authors would like to thank the reviewer for his thoughtful and useful comments on our paper. We have considered all the suggestions. Our point-by-point responses (R) to comments and questions (Q) are detailed below:

Q1:

This paper present a nonlinear phase coupling analysis of focusing wave groups propagating over a slope. Focusing wave groups with three different spectra , Gaussian, P-M, JONSWAP were generated in a wave flume. But, only three wave groups are

discussed. Actually, the topic of this study is not new. Spectral and bi-spectral analysis of irregular waves over shoal have been presented previously. I think the new aspect of this paper is presenting bispectral analysis for waves with different spectral types. My comments are outlined as follow.

R1:

We would like to thank the reviewer for the comment. One wave train from each spectrum was selected to be detailed in the text in order to avoid, as much as possible, unduly redundant results and text. Each selected wave train has the strongest steepness in its spectrum type. An important objective of this study was to investigate strong steepness wave trains. The spatial evolution of bicoherence of the other four wave trains are summarised in the last figure (Figure 11) in order to investigate the impact of the steepness on the spatial evolution of the phase coupling.

A new paragraph has been added in the conclusion (Line 336-344) in order to summarise the new aspect of this paper, which is as mentioned by the reviewer "presenting bispectral analysis for waves with different spectral types".

Q2:

Page 2, lines 47-47. The authors argues that "It is important to mention here that resonant interactions are not easily achieved in unidirectional wave train propagation since the resonant conditions cannot not be satisfied in a small area. ", I think it is better to specify that this is only true in shallow water. As for deep or intermediate water depth, resonant interactions is very strong in unidirectional wave trains.

R2:

We completely agree with the reviewer. We have added 'shallow water regions' to this sentence. Line 47.

Q3:

The reference Guohai et al. 2008 should be Dong et al. (2008).

R3:

I have corrected this error throughout the manuscript (Lines 35, 51, 62, 142, 143, 151, 154, 308 and 327).

Q4:

Fig. 3, There is a mistake in captain: JONSWAP (gamma = 7) (Test 7)

R4:

I have corrected this error and changed $\gamma$ = 3.3 to $\gamma$ = 7 (Line 199). Q5:

Where is the incipient breaking points and spatial range should be presented in the text.

R5:

I would like to thank the reviewer for this comment. The presence of the incipient breaking points and spatial range in the text is crucial for a better understanding and have been added in lines: 186, 186, 187, 202, 237, 247, 259, 284 and 285.

Q6

The authors should concentrate on the analysis of difference biphase coupling for different spectra?

R6

I agree with the reviewer. Information concerning the phase coherence can be obtained by calculating the biphase parameter ($\beta$ (a1, a2)). It will be interesting to quantitatively measure the deviation of biphase values between primary waves/higher harmonics and to analyse their spatial evolution through different spectra to distinguish differences. This text was added to the manuscript (conclusion and perspectives) (Line 353-355)

In this study, we wanted to exclusively study non-linear wave interactions through the spatial evolution of the bicoherence. In our future work, we will continue to examine this issue on selected high nonlinear wave groups by investigating accurately the spatial evolution of the biphase. It would also be useful to compare it to the results found in Ma et al. (2010).

Please also note the supplement to this comment:
https://nhess.copernicus.org/preprints/nhess-2020-209/nhess-2020-209-AC2-supplement.pdf

---

## Author Response (AR2)

The authors would like to thank the editor for his thoughtful and useful comments on our paper. We have considered all the suggestions. Our point-by-point responses (R) to comments and questions (Q) are detailed below:

Q1:

Your manuscript is greatly improved after the revision, however I am missing strong connection of your research to natural hazards. I suggest *to add a separate paragraph in the Conclusions, discussing how your results can be applied to shallow water freak waves, and may be even to tsunami of volcanic origin*. Please, think, how your findings can impact their monitoring, assessment, etc. This will make your manuscript much more attractive to NHESS readers. **I suggest also to include 1-2 extra lines in the abstract as well.**

R1:

*We would like to thank the editor for his remark. We added a separate paragraph at the end of Conclusions and perspectives (Line 353-362).*

Shallow water extreme waves are a major threat to offshore structures and ships. Findings in this study would improve our understanding of the propagation and breaking of extreme wave trains and help engineers in monitoring the wave propagation in coastal regions. The experimentally measured wave signals are highly nonlinear, unsteady and nonstationary. Consequently, the application of time-localized bicoherence analysis is shown to be a powerful approach. This study shows that an extreme wave can be readily identified from the wavelet-based bicoherence spectrum, in which strong energy is transferred to high-frequency components during the shoaling process. Such a detailed examination of individual nonlinear interactions is useful for practical applications such as investigating nonlinear responses of high frequency loads observed in severe sea conditions (e.g. springing and ringing which are excited by the sum-frequency components of irregular waves). By identifying which wave components are the most involved in the propagation process, this study may provide a complementary approach to existing experimental and field studies for determining extreme wave group runup and overtopping.

*We included 1 extra line in the abstract as well (Line 7-8).*

Prediction of non-linear wave-wave interactions is crucial in assessing the propagation of shallow water extreme waves in coastal regions.

Q2: ***Minor comment for Line 14 of the abstract: unfinished statement "approaches 1 just"***

R2: *We completely agree with the editor. This sentence have been reviewed (Line 14-16).*

[revised manuscript text omitted]